# Fine-Tuning Convolutional Neural Networks for COVID-19 Detection from Chest X-ray Images

**DOI:** 10.3390/diagnostics11101887

**Published:** 2021-10-13

**Authors:** Wentao Zhao, Wei Jiang, Xinguo Qiu

**Affiliations:** 1College of Mechanical Engineering, Zhejiang University of Technology, Hangzhou 310023, China; zhaowentao@zime.edu.cn (W.Z.); xgqiu@zjut.edu.cn (X.Q.); 2School of Intelligent Transportation, Zhejiang Institute of Mechanical & Electrical Engineering, Hangzhou 310053, China

**Keywords:** convolutional neural network, COVID-19, X-ray images, transfer learning, Grad-CAM

## Abstract

As the COVID-19 pandemic continues to ravage the world, the use of chest X-ray (CXR) images as a complementary screening strategy to reverse transcription-polymerase chain reaction (RT-PCR) testing continues to grow owing to its routine clinical application to respiratory diseases. We performed extensive convolutional neural network (CNN) fine-tuning experiments and identified that models pretrained on larger out-of-domain datasets show an improved performance. This suggests that a priori knowledge of models from out-of-field training should also apply to X-ray images. With appropriate hyperparameters selection, we found that higher resolution images carry more clinical information, and the use of mixup in training improved the performance of the model. The experimental showed that our proposed transfer learning present state-of-the-art results. Furthermore, we evaluated the performance of our model with a small amount of downstream training data and found that the model still performed well in COVID-19 identification. We also explored the mechanism of model detection using a gradient-weighted class activation mapping (Grad-CAM) method for CXR imaging to interpret the detection of radiology images. The results helped us understand how the model detects COVID-19, which can be used to discover new visual features and assist radiologists in screening.

## 1. Introduction

The novel coronavirus was reported in Wuhan, China in December 2019. Soon after its discovery, the virus was termed severe acute respiratory syndrome coronavirus 2 (SARS-CoV-2), and the disease it caused was called the Coronavirus disease 2019 (COVID-19) by the World Health Organization [1]. The most common and specific symptoms are coughing and fever with some other nonspecific symptoms such as headache, fatigue, and dyspnea [2]. Early detection of positive COVID-19 cases is a key factor in slowing the growth rate of the pandemic.

Currently, the predominant screening method is the reverse transcriptase-polymerase chain reaction (RT-PCR) testing [3], which detects SARS-CoV-2 by collecting respiratory specimens from nasopharyngeal or oropharyngeal swabs. However, recent studies showed that the sensitivity of RT-PCR testing is relatively low and that the task is both time-consuming and laborious [4,5]. False-positive SARS-CoV-2 PCR test results do occur in clinical settings and it is particularly a problem in low-prevalence screening situations where the priori probability of a positive test is low [6]. Alternatively, symptoms caused by COVID-19 can be detected through chest radiography imaging [7], including computed tomography (CT) and chest X-ray (CXR) [8], by a radiologist. In medical imaging, CT images usually have more details whereas CXR images are easier to acquire. CT scanners require expensive equipment and maintenance and are often not portable. Based on examination performed in an isolation room, it was found that the portability of the CXR system reduces the risk of COVID-19 transmission in a way that is not possible with a fixed CT scanner. Importantly, CXR imaging is a routine clinical procedure for respiratory disease [9] and is therefore often performed simultaneously with viral testing to reduce patient volume. CXR radiographic screenings are a complementary screening method to RT-PCR [10]; this method is being increasingly used in clinical institutes worldwide. It was suggested that CXR imaging may be useful for patients who return to the emergency room with worsening symptoms after an initial negative RT-PCR test result [11]. In addition, portable CXR was found to be highly valuable for critically ill COVID-19 patients [12]. Wong et al. [13] showed chest radiograph abnormalities in SARS-CoV-2-positive cases, specifically, consolidation, which was the most common finding followed by ground-glass opacities. Peripheral distribution and inferior zone distribution were the most common locations of abnormalities, with most having bilateral involvement. The interpretation of various X-ray images is usually done manually by radiologists. If we can use deep learning to automatically detect X-ray images, it can assist radiologists in COVID-19 detection. For this purpose, we built a COVID-19 detection system based on a convolutional neural network using X-ray images. We used machine learning to detect CXR images from routine clinical procedures to enable large-scale patient screening. This makes large-scale screening of potential patients a reality, as it is impractical to screen all CXR images in a routine clinical procedure using a manual approach. By changing the transfer learning source domain, we hoped to explore how different task- and data-driven weight initializations affected the performance of the model for COVID-19 detection. Furthermore, we evaluated our model performance with a smaller amount of downstream training data and found that the model still performed well in COVID-19 identification. Finally, we explored the mechanism of model detection using the gradient-weighted class activation mapping (Grad-CAM) method to assist radiologists in further analysis.

## 2. Related Work

Deep learning techniques can explain complex problems by learning from simple descriptions. Recently, deep learning methods coupled with chest radiological imaging were widely used for COVID-19 detection to accelerate the diagnosis of COVID-19. Some studies first applied a deep network to extract image features; they then applied machine learning to detect COVID-19. Tuncer et al. [14] used the residual exemplar local binary pattern (ResExLBP) to extract features and selected them based on the iterative ReliefF. The SVM classifier achieved a perfect rate (100%) for COVID-19 detection using X-ray images. Ismael and Şengür [15] introduced a deep-learning-based approach for the detection of COVID-19 based on CXR images. Pretrained deep CNN models were used for deep feature extraction; then, the SVM classifier was used to classify the deep features. The deep features extracted from the ResNet50 and the SVM with a linear kernel function produced the highest score (94.7%) amongst all the obtained results. Sethy et al. [16] introduced a deep feature plus SVM based method for detecting COVID-19 using X-ray images. Features were extracted from the fully connected layer of the CNN model, and then fed to the SVM for classification. The method achieved good robustness when compared to that of the transfer learning approach in small datasets. Elkorany and Elsharkawy [17] proposed a medical system called “COVIDetection-Net” for the automatic detection of COVID-19 infection from chest radiography images. The system used ShuffleNet and SqueezeNet architectures to extract deep learned features and used multiclass SVM for detection.

CNN is considered to be one of the most effective deep learning methods for accurately analyzing medical images. The research is mainly done through a tailored CNN, data augmentation, feature fusion, and transfer learning. A tailed CNN architecture means that the network is tailed to detect COVID-19 with a specific dataset. Mukherjee et al. [18] proposed a lightweight CNN-tailored shallow architecture that could automatically detect COVID-19 using CXRs. The proposed model achieved the highest possible accuracy when compared to that of other deep learning models, and the results stated that the proposed CNN could be advantageous for mass screening. Wang et al. [19] introduced COVID-Net, a deep CNN design tailored to detect COVID-19 cases from CXR images. They also used an explainability method to investigate how a model made predictions. They claimed COVID-Net to be one of the first open-source network designs for COVID-19 detection, achieving 93.3% on the self-built dataset COVIDx. Furthermore, they investigated how COVID-Net made predictions that could assist clinicians in improved screening. Hussain et al. [20] proposed a novel CNN model called CoroDet for automatic detection of COVID-19 using CXR and CT images. The results showed its superiority over the existing methods. Pavlova et al. [21] built COVID-Net CXR-2 to be tailored for COVID-19 case detection from CXR images. It used an interpretability-driven approach, which found that the critical factors utilized by the model were consistent with the interpretations of the radiologist.

Data augmentations have potential because the COVID-19 CXR data are very limited. Waheed et al. [22] presented a CovidGAN generation of synthetic CXR images to augment the training dataset to enhance the performance of the CNN. By adding synthetic images, the CNN model accuracy improved by 10%. Nishio et al. [23] proposed a computer-aided diagnosis system, which used VGG16 as a pretrained model and combined conventional methods and mixup to obtain a data augmentation method. They achieved 83.6% accuracy between healthy, non-COVID-19 pneumonia and COVID-19 pneumonia from CXR images. Monshi et al. [24] optimized the data augmentation and hyperparameters for detecting COVID-19 from CXRs. They proposed a CovidXrayNet model that was based on EfficientNet-B0 with an optimization strategy. The model achieved an optimal accuracy of 95.82% on the COVIDx dataset. Feature fusion means incorporating expert knowledge into automatic feature models. Rajpal et al. [25] proposed a novel classification framework, which combined a set of handpicked features with those from the CNN. The results showed the proposed framework outperformed the others in accuracy and sensitivity.

Transfer learning is a strategy used by a CNN to mine knowledge from a given data being transferred to another related task involving new data [26,27,28]. These methods train the weights of the network on large datasets and fine-tune the weights of the pretrained network using small datasets. Because only a limited amount of data is present in the current CXR datasets, the use of transfer learning is extremely important for effective COVID-19 detection [29]. With transfer learning, Apostolopoulos and Mpesiana [30] detected various abnormalities from small X-ray images; the results showed that deep learning with X-ray imaging utilizing transfer learning could successfully extract biomarkers related to the COVID-19 disease. Narayan Das et al. [31] developed a transfer learning-based approach for COVID-19 detection from X-ray images using the Xception model. The performance of the proposed model was significantly better than that of the existing models. Farooq and Hafeez [32] presented a three-step technique to fine-tune the pretrained ResNet-50 architecture to improve the performance of the model. This approach, along with the automatic learning rate selection, allowed the model to achieve an accuracy score of 96.23% on the COVIDx dataset of only 41 epochs. Nayak et al. [33] proposed a deep learning architecture to detect COVID-19 using X-ray images. Eight CNN models were used based on the concept of transfer learning; the best performance was 98.33%, which was obtained by ResNet-34. Khan et al. [34] proposed CoroNet, which was based on Xception pretrained on the ImageNet dataset to detect COVID-19 using CXR images. The results showed that CoroNet obtained 89.6% and 95% in tasks with four and three categories, respectively. From the aforementioned studies, for COVID-19 detection from CXR images, transfer learning is generally performed using the ILSVRC-2012 version of ImageNet [35] for model pre-training, and the impact of pretraining on the performance of the model is based on the adequacy of training and the size of the dataset. To fill this gap regarding pretraining only with ILSVRC-2012, we pretrained the model on a larger dataset, ImageNet-21k [36] to further enhance the pretraining performance. Furthermore, we evaluated the influence of hyperparameters and downstream dataset size on the effectiveness of transfer learning to obtain a better transfer learning method and improve the model performance for COVID-19 detection.

## 3. Materials and Methods

### 3.1. Dataset

As the COVID-19 pandemic spreads globally, access to first-hand CXR images and clinical information is becoming critical to guide clinical decision-making, deepen medical imaging understanding of viral infection patterns, and provide a systematic model for early diagnosis and timely implementation of medical interventions. The key pathway is to establish a comprehensive dataset of open access X-ray images and clinical symptoms to facilitate the global fight against the COVID-19 outbreak. To date, several datasets were created for researchers, physicians, and data scientists to conduct research on COVID-19 detection. Currently, the largest and most diverse benchmark dataset among researchers is the COVIDx CXR-2 [21]; example images are shown in Figure 1. The dataset was curated from a cross-national cohort of nearly 15,000 patients from at least 51 countries. The distribution of the CXR images in the dataset for SARS-Cov-2 negative and positive cases and the distribution of the patients is shown in Figure 2. A total of 2358 images were available for 1505 SARS-CoV-2-positive patients, and 13,993 images were available for 13,851 SARS-CoV-2-negative patients. Of these, the negative cases included no pneumonia and non-SARS-CoV-2 pneumonia images; specifically, with 8418 no pneumonia images from 8300 patients and 5575 non-SARS-CoV-2 pneumonia images from 5551 patients. The uneven distribution of samples in the training set enhances the difficulty of the model training because the number of SARS-CoV-2-positive images in the data is far less than the number of SARS-CoV-2-negative images. The test set consists of 200 SARS-CoV-2-positive images from 178 patients and 200 SARS-CoV-2-negative images from 100 no pneumonia and 100 non-SARS-CoV-2 pneumonia patient cases. We believed that the test results obtained from testing on a balanced test set are more representative. These test images were randomly selected from an international cohort of patients curated by the Radiological Society of North America [37,38], which was collected and professionally annotated by an international panel of scientists and radiologists from different institutes around the world. The test set was selected in such a way to ensure that there was no patient overlap between the training and test sets.

### 3.2. Model Selection

As presented in Section 2, Wang et al. [19] and Pavlova et al. [21] used machine-driven designs to detect COVID-19 cases from CXR images. In our work, we focused on the effect of different pretraining parameters on model training. We used ResNet-50 × 1 [39] with a vanilla ResNet-v2 architecture [40]. The ResNet-v2 model structure and the original structure are shown in Figure 3. Model performance is effectively improved by rearranging the activation functions (ReLU and BN). Firstly, using identity mapping in gray arrows eases optimization. Secondly, using BN as preactivation enhances regularization of the models. When the number of images on each accelerator was too low, the performance of the batch normalization [41] (BN) degraded [42], and upon accumulating the BN statistics of all the accelerators, the large batch computation jeopardized the generalization and caused significant delays [43]. Therefore, we substituted group normalization [44] for BN and used weight standardization [45] in all the convolutional layers. To explore how transfer learning affected the performance of CXR-based COVID-19 detection, we used ILSVRC-2012 and ImageNet-21k to pretrain the parameters of the model and fine-tuned it according to the COVID-19 detection task.

### 3.3. Downstream Fine-Tuning

To reduce the adaptation cost per task, we did not perform any hyperparameters scanning downstream. We explored the effects of different schedules, resolutions, and usage of the mixup on the performance of the model. For each iteration, we randomly selected *b* X-ray images to calculate the gradient and update the network parameters. We implemented a batch rebalancing strategy to promote a balance of positive and negative SARS-CoV-2 cases in the same batch. Unlike the previous standard training procedure, we did not limit the epochs but rather limited the schedule length. In choosing the hyperparameters, we used the stochastic gradient descent [46] with an initial learning rate of 0.003, momentum 0.9, and batch size 64. We used random crops and horizontal flips, followed by the normalization of the training data. For the test data, we used random crops followed by normalization. For the schedule length, we first conducted a warm-up [47] for the learning rate, and then reduced the learning rate three times at a rate of 10× during the entire training process. The details of the hyperparameters of the schedule length and the random crops strategy are described in detail in Section 4.1. Finally, we used the mixup [48] (Equation (Equation 1)) with λ=0.1 as set in Kolesnikov et al. [49] for data augmentation.
(1)$$\left\{\begin{array}{c} \tilde{x}=\lambda x_{i}+\left(1-\lambda \right)x_{j}\\ \tilde{y}=\lambda y_{i}+\left(1-\lambda \right)y_{j} \end{array}\right.$$

Here, $x_{i}$ and $x_{j}$ are the initial vectors while $y_{i}$ and $y_{j}$ are the raw labels. Through mixup, we obtained new vectors $\tilde{x}$ and labels $\tilde{y}$ as the new input vectors and labels. The loss function is defined using the binary cross entropy loss function (Equation (Equation 2)).
(2)$$\begin{array}{c} \ell \left(x,y\right)=\mathrm{mean}\left({\left\{l_{1},\ldots ,l_{N}\right\}}^{⊤}\right)\\ \;l_{n}=-\left[y_{n}\cdot \log\sigma \left(x_{n}\right)+\left(1-y_{n}\right)\cdot \log\left(1-\sigma \left(x_{n}\right)\right)\right] \end{array}$$
where *N* is the batch size, σ is the sigmoid activation function, and $x_{n}$ and $y_{n}$ are the output vectors of the model and label, respectively. We used the mean function to calculate the loss.

## 4. Results and Discussions

### 4.1. Hyperparameters Sensitivity

In this section, we explore the impact of various hyperparameters on the performance of the model, in particular: the schedule length, image resolution, and use of the mixup. We used a combination of four overall schedule length selections and random crops strategies (image resolution). For the random crops strategy of CXR images, we adopted the settings (160, 128), (256, 224), (448, 384), and (512, 480), where the first value in each pair of values indicates the scale of adjustment during training and the second value indicates the scale of random cropping during training and the scale of adjustment during testing. Regarding the schedule length, we used [100, 200, 300, 400, 500], [500, 1500, 3000, 4500, 10,000], and [500, 6000, 12,000, 18,000, 20,000]. The first parameter indicates the number of steps in the warm-up step, the last is the end step, and the rest are the step nodes where the learning rate decays by a factor of 10. Figure 4 displays the test accuracy for different resolutions and schedule lengths with and without the mixup. BiT-M is trained on the full ImageNet-21K dataset, a public dataset containing 14.2 million images, and a WordNet hierarchy organized in 21K classes. Images may contain multiple labels. BiT-S is trained on the ILSVRC-2012 variant of the ImageNet, which contains 1.28 million images and 1000 classes. Each image has a single label. All pretrained models are from Kolesnikov et al. [49]. The results show that the higher the image resolution, the higher the detection accuracy of COVID-19; therefore, clear CXR images contain more diagnostic clinical information. This illustrates that high-resolution images carry a large number of detailed features that facilitate the model to learn local information. A longer schedule length can also improve the accuracy; however, the results are less visible when it exceeds 10,000. It means a lengthy schedule can cause overfitting of the model and may cost more training time. Owing to the lack of data set samples, especially for SARS-CoV-2-positive cases, the use of the mixup significantly improves the performance of the model, even frequently surpassing the gains for the pretraining model. This indicates that mixup enhances the richness of the training data, thus enhancing the model generalizability. By testing the performance of the hyper-parameters on the model, we used (512, 480) as a random crops strategy and [500, 1500, 3000, 4500, 10,000] as the schedule length with mixup to test the performance of the model in Section 4.2.

### 4.2. Test Performance

We trained the model with different initializations of parameters and used the training settings described in Section 3.3. To classify the model output after the sigmoid function, we fixed the threshold value to 0.5. If the output was less than or equal to 0.5, the predicted result was negative; if it was greater than 0.5, the result was positive. The quantitative performance results were compared to those of the current novel methods; the results are presented in Table 1. Bit-M and Bit-S were introduced in Section 4.1, and random means random initialization of model parameters. Only 2358 positive SARS-CoV-2 images were used in our training dataset, as compared to 5210 positive SARS-CoV-2 images used in a previous study [21], making it more difficult to train our models. Even in this case, the performance of BiT-M (at 96.5%) remained on par with COVID-Net CXR-2 [21] (at 96.3%). Bit-S achieved a 0.8% higher test accuracy (at 94.8%) than COVID-Net (at 94.0%), and random achieved a 2.3% higher test accuracy (at 92.8%) than ResNet-50 [40] (at 90.5%). With random initialization, our model outperformed ResNet-50, indicating that the expressiveness of the model effectively improved through a series of model architecture improvements; these were illustrated in Section 3.2. By changing the pretraining parameters of the models, our models significantly improved in terms of their sensitivity (from 85.5% to 93.0%) and negative predictive value (NPV) (from 87.3% to 93.5%). High-sensitivity and -NPV ensured that few false negative cases led to miss diagnoses during the screening process. Too many false negative cases prevents the system from effectively screening our patients. This means that patients return to the community, resulting in community transmission. From a clinical perspective, this transmission is important to control if we are to regulate the spread of the SARS-CoV-2 virus during the ongoing COVID-19 pandemic. For both specificity and positive predictive value (PPV), all our models obtained 100%. This indicates that there are no false positive cases, that is, patients identified as positive are all true positive cases. This has the potential to decrease the burden on the healthcare system. In addition, identification of the true negative cases as false positives greatly increases the psychological burden of the patients. From the AUC, which is not related to threshold, we found that the proposed model also has a good performance. We calculated the test performance of different pretrained models for every 100 steps, as presented in Figure 5. Owing to the small dataset and the large learning rate used at the initial steps of the training, variations in the test performances at the beginning were significant and the pretrained models showed a highly robust performance. The model pretrained on ImageNet-21K (BIT-M) exhibited a better performance in the evaluation with the test set at the latter stages than the one pretrained on the ILSVRC-2012 (BIT-S) or random initializations (Random). This result implies that model pretraining is crucial for ensuring desirable model performance and that pretraining on larger datasets has a stronger generalization. Our proposed model enhances the model performance by improving the model structure. More importantly, by pretraining on a larger pretraining set, more prior knowledge is learned, which significantly improves performance in downstream tasks. Finally, Figure 6 shows the performance of Bit-M in more detail through the confusion matrix [50].

### 4.3. Size of Labeled Training Data

A common problem in medical image analysis is that most CXR images used for diagnostic purposes are not publicly accessible owing to privacy issues. The lack of open-source datasets for CXR images presents a significant obstacle to the development of more advanced deep learning techniques to better detect COVID-19. To better match the real situation where data are harder to obtain, we down-sample the data, which means we randomly select a certain number of images from each category to test the performance of the models. For each category, we randomly chose 32, 64, 128, 256, and 512 samples for training and tested the trained models to observe their recognition rate with the test set. The results of these tests are presented in Figure 7. The histogram on the right shows the accuracy of the Imagenet21k pretrained model, COVID-Net, and ResNet-50 using the entire training set, respectively. We found that the pretrained models consistently outperformed the case of random initialization of parameters for different training dataset sizes with down-sampling. This illustrates the importance of using transfer learning during training. In most cases, the Bit-M outperformed the Bit-S, which indicates that the model pretrained on a larger out-of-domain dataset has a stronger generalization. We also identified an interesting phenomenon wherein the performance of the models peaked when each class took 64 samples. This indicates that a slight over-fitting effect occurred when 128 samples were taken per class. Subsequently, the performance of the models continuously improved with the increase in the size of the training dataset. When selecting 512 samples per class, Bit-M (94.75%) outperformed COVID-Net (94.0%). This supports that our models with transfer learning can still work well on limited datasets. The prior knowledge obtained by pretraining on large datasets ensures excellent performance despite the limited training data.

### 4.4. Qualitative Analysis

Although the effectiveness of the proposed method in detecting COVID-19 from CXR images is evident from the classification accuracy that was achieved, it is necessary to compare the results to clinical evidence for the former to be useful in clinical practice. To this end, we used the Grad-CAM [51] visualization technique to explore areas of concern for the model in COVID-19 detection and better understand the areas in the CXR images that are critical for the diagnostic accuracy of the model. Grad-CAM uses the gradient of any target concept and places it in the final convolution layer to generate a coarse localization map that highlights the key regions in the image to predict the required concepts. It can be used to help users establish appropriate trust in predictions based on deep networks. As demonstrated in Figure 8, we first randomly selected two SARS-CoV-2-negative images and two SARS-CoV-2-positive images and enlarged them to 480 × 480 pixels before using Grad-CAM for visual explanation. The model we used was Bit-M; all the predictions performed by the model in Figure 8 are the same as the actual detection results. We found that for the SARS-CoV-2-positive cases, the region of interest in the chest could be well localized. However, for the SARS-CoV-2-negative cases, the model appeared more interested in the edges; the reason behind this deserves further investigation. The discovery of the underlying reason will help explain new visual indicators to help radiologists detect COVID-19 for manual screening based on CXR images.

## 5. Conclusions

Our study applied transfer learning on COVID-19 using chest X-ray (CXR) images and discussed the impact of various initializations of parameters on model performance. The results showed that the model pretrained on ImageNet21K exhibited a stronger generalization. The proposed model (Bit-M) provided an accuracy of 96.5% for detecting COVID-19 cases from CXR images. The generalization ability of the model was effectively enhanced through transfer learning, especially by pretraining on a larger dataset. Specifically, the metrics of sensitivity and negative predictive value (NPV) effectively increased, which meant a significant reduction in the number of false positive cases. An excessive number of false negative cases enhanced the risk of community transmission. Our models achieved 100% in the positive predictive value (PPV) and specificity metrics, which meant that there were zero false positive cases, i.e., patients that were predicted as positive were actually true positives; this has the potential to greatly reduce the burden on the healthcare system, especially in the case of a pandemic outbreak. In addition, to model the real situation better, where medical datasets are more difficult to access because of privacy and other issues, we examined the performance of the models with different initialization parameters under limited data conditions and found that models still performed adequately. This suggests that our models are still applicable to limited data, which better resembles the real situation since large and diverse datasets may not be readily available. Finally, we used Grad-CAM visualization technique to explore the mechanisms associated with the model for COVID-19 detection from CXR images; thus, making the proposed model more interpretable and explainable. We found that for positive cases, the model focused more on the chest region, whereas for negative cases, the model focused more on the edge. The reason behind the model focusing on these different regions of interest should be explored further. Furthermore, we also must develop more effective visualization techniques for medical image characteristics to further understand the model detection mechanism. In this experiment, we used a relatively small test dataset to make a fair comparison with previous studies, causing a large oscillation in the performance of the models on the test dataset during training. In the future, we will refine the test dataset to test the performance of models more fairly. Although the system is well studied for performance on public datasets, the work is still in the theoretical research stage and the models were not practically validated in clinical routine. It is not a production-ready solution, and we will use machine learning as a complement to the COVID swab test. Therefore, we will test our system in a clinical routine and communicate with physicians to understand how they use it and engage their opinions on the models. This will allow us to further refine the models in future works. Deep learning techniques should be considered when creating information packages by organizations for the procedures where shared decision making [52] is used as a tool to help health care professionals effectively communicate all relevant information to the patient or their next of kin to reach an informed decision.

## Figures and Tables

**Figure 1 diagnostics-11-01887-f001:**
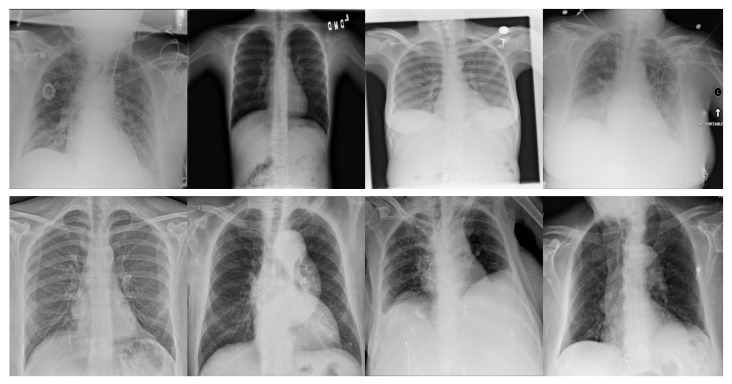
Example chest X-ray images from COVIDx CXR-2. First row is SARS-CoV-2-negative patient cases, and second row is SARS-CoV-2-positive patient cases.

**Figure 2 diagnostics-11-01887-f002:**
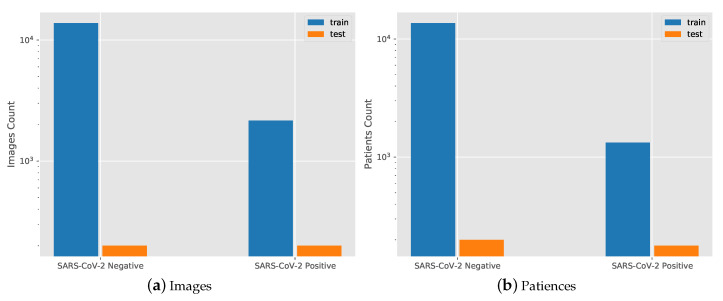
Chest X-ray image and patient distribution of SARS-CoV-2-negative and SARS-CoV-2-positive cases. (**a**) Images distribution. (**b**) Patients distribution.

**Figure 3 diagnostics-11-01887-f003:**
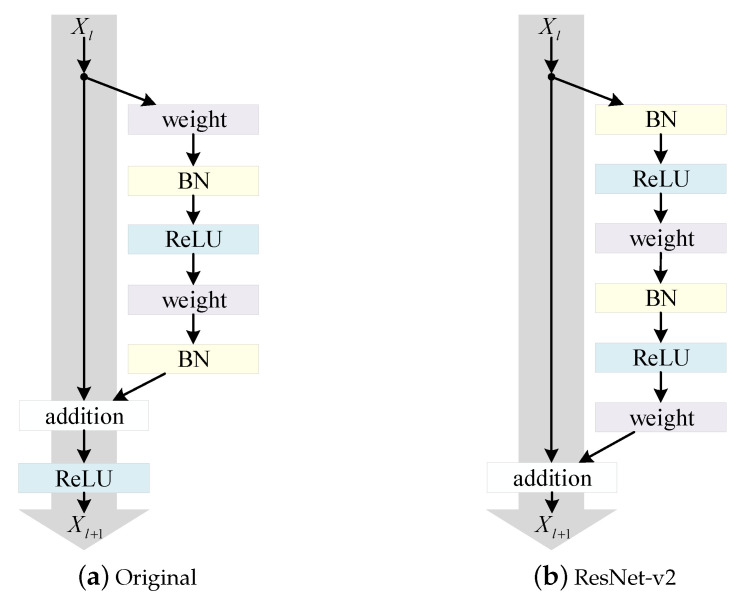
(**a**) Original Residual Unit in He et al. [39]; (**b**) ResNet-v2 Unit [40]. Gray arrows indicate easiest paths for information dissemination.

**Figure 4 diagnostics-11-01887-f004:**
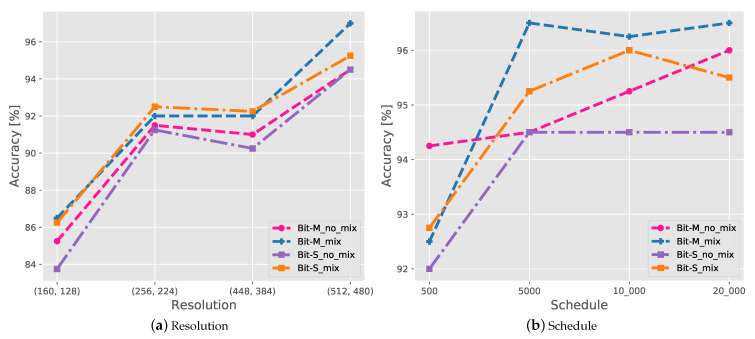
Test accuracy of COVIDx CXR-2 with various hyperparameters. (**a**) Resolution. (**b**) Schedule.

**Figure 5 diagnostics-11-01887-f005:**
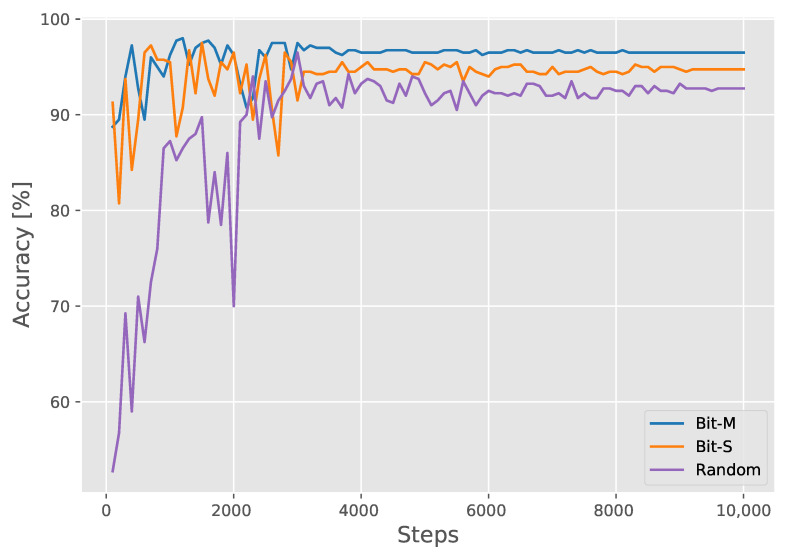
Test accuracy curves for models with different initialization parameters.

**Figure 6 diagnostics-11-01887-f006:**
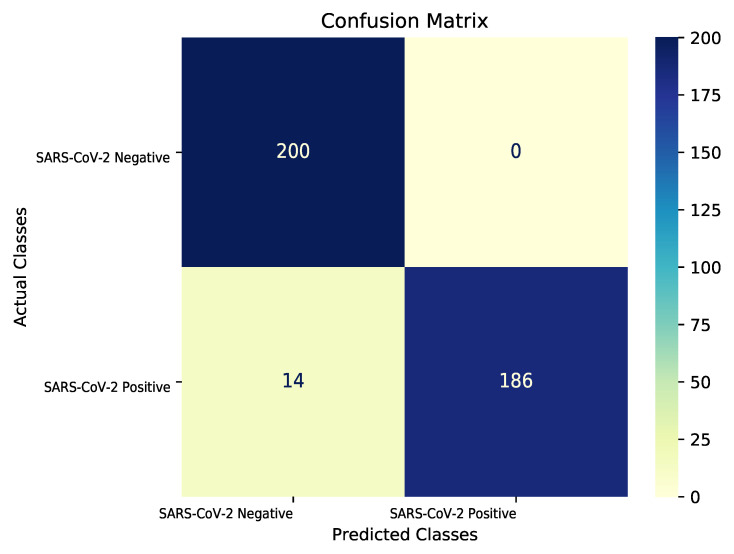
Confusion matrix for COVID-19 detection using Bit-M fine-tuning model. Color bar indicates intensity of normalization.

**Figure 7 diagnostics-11-01887-f007:**
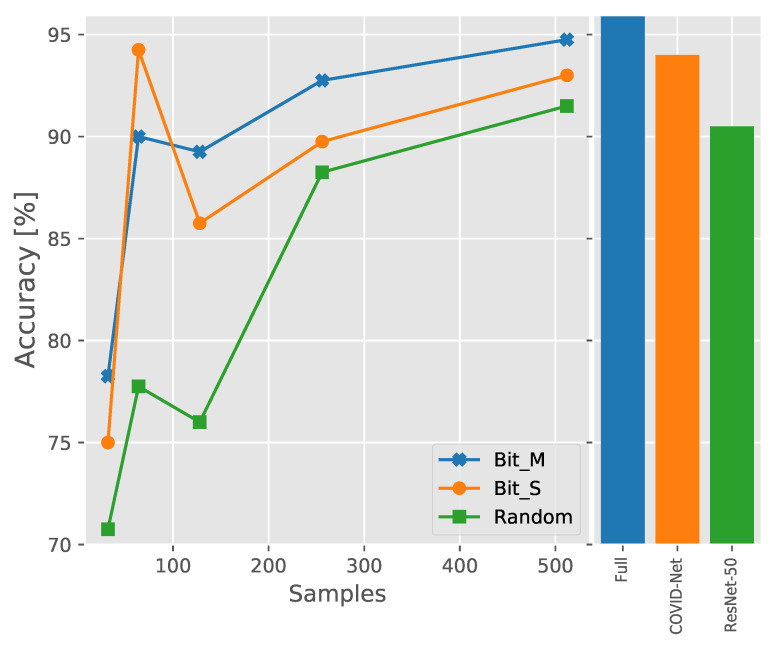
Effect of number of images per category in training set on model performance.

**Figure 8 diagnostics-11-01887-f008:**
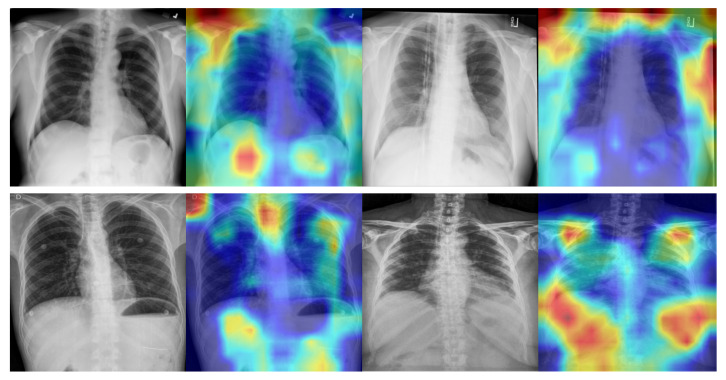
Grad-CAM visualization of Bit-M. First row is SARS-CoV-2-negative patient cases; second row is SARS-CoV-2-positive patient cases. Red regions correspond to a high score for class. All models of chest X-ray images yielded accurate predictions.

**Table 1 diagnostics-11-01887-t001:** Accuracy, Sensitivity, PPV, Specificity, and NPV of our different pretraining models on test data from COVIDx CXR-2 dataset. Best results highlighted in bold.

Nerwork	Accuracy (%)	Sensitivity (%)	PPV (%)	Specificity (%)	NPV (%)	AUC (%)
ResNet-50 [40]	90.5	88.5	92.2	-	-	-
COVID-Net [19]	94.0	93.5	100.0	-	-	-
COVID-Net CXR-2 [21]	96.3	**95.5**	97.0	97.0	**95.6**	-
Random	92.8	85.5	100.0	100.0	87.3	99.5
Bit-S	94.8	89.5	100.0	100.0	90.5	99.5
Bit-M	**96.5**	93.0	**100.0**	**100.0**	93.5	99.4

## Data Availability

Publicly available datasets were analyzed in this study. This data can be found here: https://www.kaggle.com/andyczhao/covidx-cxr2 (accessed on 12 June 2021).

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
