# Peer review of "Fine-Tuning Convolutional Neural Networks for COVID-19 Detection from Chest X-ray Images"

_diagnostics, 2021, doi:10.3390/diagnostics11101887_

Round 1
Reviewer 1 Report
Dear Authors,
This is a good paper that has only a few concerns to fix.
You demonstrate how to improve the ML diagnostic tools and diagnosis of COVID-19.
Consider discussing briefly these points.
Introduction:
It should better explain the problem related to the swab test and false positive rate and the positive predicting value of the swab test.
You demonstrate how to improve the ML diagnostic tools and diagnosis of COVID-19.
Discussion:
Economic points and ML equipment. Many hospitals have old technologies.
Clarify the objective of ML with a high predicting value. What is the real clinical impact? Why should Stakeholders consider adopting these tools?
So how will you use machine learning? Do you substitute the COVID swab test in the likelihood scenario without contact tracing? It could be applicable in a setting where most of the population is vaccinated.
All this has an important implication to the patient, the measure to apply with him, and how to manage the healthcare policy and shared decision-making problem (evaluate to discuss this last point briefly, and I recommend this paper also. Maybe you should consider discussing this point briefly. By the way, I understood that the main objective is demonstrating an improvement in technical problems, and you have done it.
"Overview of the main challenges in shared decision making in a multicultural and diverse society in the intensive and critical care setting. J Eval Clin Pract. 2020"
In many conferences or academic speeches, many people underline the importance of ML, but often only people that know the topic understand the importance of advantages of all of this. From my point of view your paper is well done, it could be improved with a brief paragraph with these points.
I hope these suggestions could be helpful.
Reviewer 2 Report
The paper introduced an interesting topic of using deep learning models and breast images to classify Covid-19. After reading the paper, I have the following comments that I think the authors need to address to make the paper better:
- For the dataset used in the paper, I would recommend the authors add more info about the data, like the patients' race, sex, age. Another question regarding to the dataset is that, why the training dataset and testing dataset are selected from two different cohorts? I feel it is not reasonable to use a test dataset that is selected from different distribution as the training dataset.
- A ResNet-50 with vanilla architecture is used in the paper, I would recommend the authors include a figure and some contents showing the exact architecture and reason why such architecture is selected.
- In 3.3., how is the mixup factor (0.1) is decided? I think the other parameters are decided based on some experiments, why we don't have experiments for this specific param?
- In the experiment, only accuracy/precision/recall are reported, such metrics are related to threshold. Want to know the threshold details for such metrics. I would also recommend the authors report the AUC as a comprehensive metric, which is not related to threshold
- In each experiment, the authors only talk about the results. I would also recommend the authors to explain the reasons behind such results. For instance, why the proposed model outperforms other models. Why changing the resolution and schedule will change the model performance. This helps the audience understand more about the authors contributions.
- In section 4.4, the authors used the Grad-CAM to understand the precision of the model, which is very interesting, but they claim the model focus more on chest area for positive cases while focus more on edges for negatives. Is there any statistical analysis to support such conclusion?
Round 2
Reviewer 2 Report
The authors have addressed the comments I have made. I think it is ready to be published.